In situ visualization of bacterial populations in coral tissues: pitfalls and solutions

Wada Naohisa 1 2 3
Pollock Frederic J. 2 3 4 5 6
Willis Bette L. 3 5 6
Ainsworth Tracy 5
Mano Nobuhiro 1
Bourne David G. david.bourne@jcu.edu.au 2 3 6
1 Department of Marine Science and Resources, College of Bioresource Science, Nihon University , Fujisawa , Kanagawa , Japan
2 Australian Institute of Marine Science , Townsville , Queensland , Australia
3 AIMS@JCU , Townsville , Queensland , Australia
4 Department of Biology, Eberly College of Science, Pennsylvania State University , University Park , PA , United States
5 ARC Centre of Excellence for Coral Reef Studies, James Cook University of North Queensland , Townsville , Queensland , Australia
6 College of Science and Engineering, James Cook University of North Queensland , Townsville , Queensland , Australia
Thompson Fabiano
Electronic publication date: 2016 Sep 20
Publication date: 2016
Volume: 4
Electronic Location ID: e2424
Received 2016 Jun 3; Accepted 2016 Aug 9
Copyright: ©2016 Wada et al.
Copyright year: 2016
Copyright holder: Wada et al.
License: This is an open access article distributed under the terms of the Creative Commons Attribution License, which permits unrestricted use, distribution, reproduction and adaptation in any medium and for any purpose provided that it is properly attributed. For attribution, the original author(s), title, publication source (PeerJ) and either DOI or URL of the article must be cited.
License URL: https://creativecommons.org/licenses/by/4.0/

Keywords: Coral, Bacteria, Fluorescence in situ hybridization, Holobiont, In situ visualization

Funding: Australian Institute of Marine Science Australian Research Council Discovery Award DP130101421 This work was funded through internal funding from the Australian Institute of Marine Science and the Australian Research Council Discovery Award DP130101421. The funders had no role in study design, data collection and analysis, decision to publish, or preparation of the manuscript.

==============================
In situ visualization of microbial communities within their natural habitats provides a powerful approach to explore complex interactions between microorganisms and their macroscopic hosts. Specifically, the application of fluorescence in situ hybridization (FISH) to simultaneously identify and visualize diverse microbial taxa associated with coral hosts, including symbiotic algae (Symbiodinium), Bacteria, Archaea, Fungi and protists, could help untangle the structure and function of these diverse taxa within the coral holobiont. However, the application of FISH approaches to coral samples is constrained by non-specific binding of targeted rRNA probes to cellular structures within the coral animal tissues (including nematocysts, spirocysts, granular gland cells within the gastrodermis and cnidoglandular bands of mesenterial filaments). This issue, combined with high auto-fluorescence of both host tissues and endosymbiotic dinoflagellates (Symbiodinium), make FISH approaches for analyses of coral tissues challenging. Here we outline the major pitfalls associated with applying FISH to coral samples and describe approaches to overcome these challenges.

Introduction

Corals form close symbiotic associations with a range of microorganisms, including dinoflagellate microalgae, Bacteria, Archaea and viruses, a consortium collectively termed the coral holobiont (Rohwer et al., 2002; Rosenberg et al., 2007). Coral-associated bacterial communities are known to contribute to coral holobiont fitness (Rosenberg et al., 2007) and disruptions in coral-associated bacterial community structure have been implicated in the onset of coral disease (Bourne et al., 2009; Bourne, Muirhead & Sato, 2011). However, the identification of specific bacterial pathogens directly responsible for disease causation has been problematic and, in many cases, causative agents have not been definitively linked with disease lesions at the cellular level (Work & Aeby, 2006; Work et al., 2008). While standardized histological approaches allow investigations of disease lesions at the cellular level, in situ hybridization (ISH) approaches allow targeted localization of specific DNA and RNA sequences at the molecular scale (Levsky & Singer, 2003). Specifically, fluorescence in situ hybridization (FISH) provides a powerful tool for simultaneous identification and visualization of bacteria within host tissues (Moter & Gobel, 2000).

The application of FISH to coral samples has been complicated by the need for time-consuming and labor-intensive processing, combined with specialized, and often expensive, microscopes and image processing software to separate probe fluorescence from high background tissue auto-fluorescence (Ainsworth et al., 2007; Pollock et al., 2011). Optimized FISH protocols have been developed to overcome some of these limitations, allowing researchers to target specific nucleic acid sequences in coral histological studies (Ainsworth et al., 2006a; Ainsworth, Hoegh-Guldberg & Leggat, 2008). Consequently, FISH approaches have become a valuable tool to elucidate how microbial communities are spatially located within both healthy and diseased coral tissues (Bythell et al., 2002; Webster et al., 2004; Lesser et al., 2004; Ainsworth et al., 2006a; Ainsworth, Hoegh-Guldberg & Leggat, 2008; Ainsworth & Hoegh-Guldberg, 2009; Apprill et al., 2009). For example, Bayer et al. (2013) recently employed FISH-based techniques to identify abundant communities of Endozoicomonas-related bacteria residing within healthy coral tissues and Neave et al. (2016) showed that Endozoicomonas species which can comprise as much as 90% of the microbiome and form cyst-like aggregations at the interface of the epidermal and gastrodermal cell layers of Stylophora pistillata. Ainsworth et al. (2015) identified a core coral microbiome using sequencing-based approaches and applied FISH techniques to localize some members of this core microbiome within microhabitats of the coral host. FISH also allows detection, identification and visualization of potential coral pathogens (Ainsworth et al., 2007), which is important for studies of disease etiology and the development of diagnostic tools for coral diseases (Pollock et al., 2011).

Autofluorescence associated with corals is the direct result of high densities of chlorophyll-containing dinoflagellates within the corals’ gastrodermal layers and an abundance of fluorescent pigments, including green fluorescent protein-like molecules within coral cell layers (Salih et al., 2000; Dove, Hoegh-Guldberg & Rangananthan, 2001; Yokouchi et al., 2003; Ainsworth et al., 2006a). Despite the demonstrated utility of direct localization and visualization of specific DNA and RNA targets within coral tissues and recent advances to overcome high levels of coral auto-fluorescence, non-specific probe binding (i.e., erroneous hybridization of FISH probes to non-target structures leading to the detection of false positives) impedes the application of ISH and FISH techniques to coral samples (Bythell et al., 2002; Ainsworth et al., 2006b; Apprill et al., 2009). Clear and consistent guidelines and methodological criteria are therefore needed to differentiate specific and non-specific probe binding. In this manuscript, we outline current barriers to the application of FISH techniques to coral samples and provide clear guidelines to help researchers and practitioners overcome these challenges.

Materials and Methods

Sample collection

Samples of healthy and white syndrome (WS) infected colonies of Acropora hyacinthus (i.e., colonies displaying diffuse, acute to sub-acute areas of tissue loss revealing white, intact skeleton; see Fig. 1) were collected from reefs near Lizard Island (14°40′S, 145°27′E) in the Northern sector of the Great Barrier Reef (GBR) on SCUBA (1–5 m depth), in September 2010, July 2011 and February 2012. At each sampling time point, ∼3 cm coral fragments were collected from each healthy colony (n = 4) and from each WS-infected colony (i.e., from the lesion-healthy tissue interface (n = 7) and from apparently healthy tissue (n = 3) approximately 10 cm away from the lesion). Coral fragments were placed in individual sterile bags underwater. Sampling was undertaken on Permit G11/34003.1 issued by the Great Barrier Marine Park Authority.

Figure 1 Characteristic field appearance of a white syndrome (WS) on a colony of the coral Acropora hyacinthus from reefs near Lizard Island (14°40′S, 145°27′E) in the Northern sector of the Great Barrier Reef (GBR): (A) WS lesion on a coral colony with numbers and arrows displaying (1) dead coral skeleton overgrown with algae, (2) recently exposed coral skeleton prior to algal overgrowth, (3) actively progressing lesion boundary where samples derived for this study were taken, (4) healthy tissue ahead of the lesion and from which samples were also derived. (B) Close up image of lesion boundary displaying diffuse, acute to sub-acute areas of tissue loss revealing white, intact skeleton.

Fixation, decalcification and sectioning

Within 15 min of collection, coral samples were placed in freshly prepared 4% paraformaldehyde (Electron Microscopy Sciences, USA), 10 mM phosphate buffered saline (PBS) solution at 4 °C. After 8–10 hr, the paraformaldehyde solution was exchanged for a 1:1 solution of 10 mM PBS and ethanol. Samples were rinsed with 10 mM PBS and then embedded in 1.5% agarose (∼55–60 °C) to maintain tissue conformation during the decalcification process. Once the agarose was set, excess agarose was removed and a small hole was punched through to the coral fragment to allow liquids to reach the sample. Agarose-embedded samples were placed in histological cassettes and decalcified in a 20% EDTA solution (pH 8.0), with the solution maintained at 4 °C and exchanged approximately every 2 days for 2–3 weeks. Following decalcification, the agarose-embedded samples were rinsed in PBS, dehydrated sequentially in a 70%, 80%, 90%, 95%, 100% and 100% ethanol series for 60 min each, then processed through three, 30 min xylene rinses and embedded in Paraplast paraffin wax. Paraffin-embedded samples were serially sectioned at 4 µm and collected on Superfrost Plus slides (Menzel, Germany). Sections were dewaxed at 60 °C prior to histological processing.

Histology

Prior to staining, one serial section from each sample was dewaxed in xylene (2 × 3 min) and rehydrated through 3 × 5 min 100% ethanol, 1 × 5 min 70% ethanol, and 1 × 2 min water washes. Hydrated sections were counterstained in Mayer’s Hematoxylin for 8 min, rinsed in tap water (1 × 20 dips), differentiated in Scott’s tap water substitute for 30 s for bluing, rinsed in water (1 × 2 min) and stained in Eosin for 3 min. Stained section were dehydrated through an ethanol series (1 × 2 min 70% ethanol and 2 × 5 min 100% ethanol) and washed in xylene (2 × 5 min). Sections were mounted in DPX mounting medium, observations were recorded using a Leica DMI 6000B light microscope (Leica, Germany) and microphotograph images were processed using the LAS imaging software (Leica, Germany).

Fluorescent in situ hybridisation (FISH)

Prior to staining, three serial tissue sections were dewaxed in xylene (2 × 3 min), dehydrated through 100% ethanol washes (3 × 5 min) and air-dried. Dried sections were washed in a 0.2 M HCl solution for 12 min and a 20 mM Tris HCl solution (pH 8.0) for 10 min at room temperature. To digest bacterial cellular membranes and allow easier probe penetration into tissues, sections were mounted in a proteinase K (50 µg/ml), 20 mM Tris HCl solution (pH 8.0) for 5 min at 37 °C and rinsed in 20 mM Tris HCl (pH 8.0) prior to probe hybridisation. Oligonucleotide probes, including a probe targeting the 16S rRNA gene (EUB338 mix: 5′-GCT GCC TCC CGT AGG AGT-3′, 5′-GCA GCC ACC CGT AGG TGT-3′, 5′-GCT GCC ACC CGT AGG TGT-3′) and a nonsense, negative control probe (NonEUB338: 5′-ACA TCC TAC GGG AGG C-3′), were labeled with the Cy3 flurochrome (Thermo Fisher Scientific, Germany) (Wallner, Amann & Beisker, 1993; Daims et al., 1999). Tissue sections were covered with hybridization buffer (30% v/v formamide, 0.9 M NaCl, 20 mM Tris–HCl (pH 8.0), 0.01% SDS), oligonucleotide probes were added to a final concentration of 25 ng µl−1, and samples were incubated at 46 °C for 1.5 h. Sections were washed in 50 ml Falcon tubes containing preheated wash buffer (0.112 M NaCl, 20 mM Tris–HCl (pH 8.0), 0.01% SDS, 5 mM EDTA) in a water bath at 48 °C for 10 min. Following washing, sections were immediately soaked in cold, filtered water for 10 s to remove excess salts, air dried and mounted in Citifluor AF1 (ProScitech, Australia). As a negative control, one serial tissue section was processed as described above, but no oligonucleotide probe was added. To detect true and false binding simultaneously, one serial section was processed with the Cy3-labelled EUB338 mix probes as described above, and a Cy5-labelled NonEUB338 probe applied to the same section. An LSM710 confocal laser scanning microscope (Carl Zeiss, Germany) combined with spectral emissions profiling was used to visualize tissue-associated, FISH-labeled bacterial communities, as described by Ainsworth et al. (2006a). Detection of the Cy3 fluorochrome label was in the emission range 519–580 nm and the target signal was recorded at 561 nm. Auto-fluorescence detection and spectral removal included emission ranges of 407–486 nm for removal of coral tissue autofluorescence and 627–704 nm for removal of dinoflagellates (Symbiodinium) autofluorescence. Micrographs of bacterial communities associated with coral tissue sections were processed using Zen 2009 software (Zeiss, Germany). The linear unmixing function of the Zen 2009 software (Zeiss, Germany) was utilized when multiple probes were visualized simultaneously.

Trouble shooting approaches

To determine if the sequence of the oligonucleotide probe contributes to non-specific binding, three probes were assessed: EUB338 mix probes, NonEUB338 and a Vib-GV (5′–AGG CCA CAA CCT CCA AGT AG-3′; Giuliano et al., 1999). To determine if the fluorochrome attached to the oligonucleotide probe affects non-specific binding, three flurochromes were assessed: Atto 647 (excitation λ = 645, emission λ = 669) Cy3 (excitation λ = 548, emission λ = 561) and Cy5 (excitation λ = 647, emission λ = 665) (i.e., Cy3-labeled EUB338 mix, Cy3-labeled NonEUB338, Cy5-labeled NonEUB338 and Atto 647-labelled Vib-GV). To assess the utility of incorporating hybridising agents in the FISH workflow to avoid non-specific binding, “Blocking Reagent” (Roche, Germany) was added to the hybridisation buffer (30% formamide, 0.9M NaCl, 20 mM Tris–HCl with adjusted pH8.0, 0.01% SDS, 10% “Blocking Reagent” with maleic acid buffer), alongside Cy3-labelled NonEUB338 probe.

Figure 2 Detection and characterization of specific and non-specific fluorescence in situ hybridization (FISH) probe binding to target bacteria (BA and Bac), granular gland cells (Gc), and spirocysts (Sp) using Cy3-labelled FISH probes (A–C, E, F, H, I) and Hematoxylin and Eosin (H&E) staining of coral tissue sections (D, G).

Non-specific binding of EUB338 (A) and nonEUB338 (B) FISH probes to granular gland cells and lack of auto-fluorescence of these structures in probe-free treatments (C) is demonstrated through serial tissue sections. Detailed granular gland cell morphology within the gastrodermis is shown in H&E stained (D) and EUB338 FISH-hybridized (E) tissue sections. Non-specific binding of EUB338 FISH probes to spirocysts within epidermal cells (F) was detected in tissue sections and was particularly prevalent in coenosarc and tentacle regions of polyps. Detailed spirocyst morphology is shown through H&E staining (G). A bacterial aggregate within the calicoblastic layer is shown near to non-specific binding signals of granular gland cells (H) in healthy coral tissues hybridized with EUB338 FISH probes. Bacteria assemblages were detected within necrotic tissues associated with WS disease using EUB338 FISH probes (I). Scale bars represent 50 µm in (A–C) and 10 µm in (D)–(I). Abbreviations: Gc, granular gland cell; Sp, spirocysts; Symb, Symbiodinium; BA, bacterial aggregation; Bac, bacterial assemblages.

Results and Discussion

The challenges of applying FISH approaches to corals

The challenge of non-specific FISH probe binding within coral tissues is clearly highlighted in FISH-labeled sections of healthy samples of Acropora hyacinthus from Lizard Island in the northern Great Barrier Reef (GBR) (Fig. 2). Under fluorescence excitation light, coral tissues targeted with probes for Bacteria (EUB338) show multiple strongly fluorescent cellular structures within anatomical features of coral polyps (e.g., mesenterial filaments, Fig. 2A), suggesting the presence of targeted bacteria within gastrodermal and epidermal cells. However, direct comparison with serial sections targeted with nonsense negative control probes (NonEUB338) (i.e., probes specifically designed to detect non-target binding) (Fig. 2B) highlights strong non-specific binding of both probes to granular gland cells within the cnidoglandular band of mesenterial filaments. Granular gland cells are secretory cells commonly distributed throughout the gastrodermis, including in regions of coenosarc, stomodeum and mesenterial filaments, where secretions are released into the corals’ coelenteron (gastrovascular cavity) to aid extracellular digestion of prey (Galloway et al., 2007). The lack of granular cell fluorescence in probe-free, negative control sections (Fig. 2C) indicate that these false positives are caused by non-specific binding of the oligonucleotide probes rather than auto-fluorescence. Granular gland cells within the gastrodermis appear as pink aggregations in hematoxylin and eosin (H&E) stained histological sections (Harrison, 1980; Fig. 2D). Due to their strong fluorescence signal in FISH images (Figs. 2A, 2B and 2E), combined with their spherical shape and similar size to bacteria (approximate diameter = 0.5 µm), non-specific binding of granular gland cells can easily be confused with true binding to small cocci bacterial cells.

Spirocysts, which are a type of cnidae comprised of a single-walled capsule containing a tightly coiled tubule bearing microtubules (Galloway et al., 2007), were another common site for non-specific FISH probe binding. Spirocysts were frequently detected in the epidermal layer and were particularly prevalent in the coenosarc and tentacles (Figs. 2F and 2G). Unlike granular gland cells, spirocysts can be easily differentiated from bacterial aggregations due to their coiled tubules (Figs. 2F and 2G). However, this distinction may not be obvious to the untrained practitioner. Non-specific binding of FISH probes to common cellular structures (granular gland cells and spirocysts, in particular) within coral tissues highlights the importance of including non-target (i.e., NonEUB338) probe controls and extensive familiarization with coral cellular structure to differentiate true positive signals from false positives.

Common approaches designed to avoid non-specific binding failed to reduce non-specific fluorescent signals generated by false probe binding to spirocysts and granular gland cells. The incorporation of alternative oligonucleotide probes, which targeted different 16S rRNA sequences with different hybridization kinetics and efficiencies (competitor probes), also yielded non-specific hybridization to granular gland cells (Fig. 2C) and spirocysts (Fig. 3). Increased probe hybridization stringencies through higher formamide concentrations in the hybridization buffers, plus higher stringency post-hybridization washing as recommended by Wallner, Amann & Beisker (1993), also failed to reduce the strong fluorescent signal associated with these cellular structures. Furthermore, the application of commercially available blocking solutions, which are specifically designed to prevent non-specific binding, also failed to prevent non-specific binding of oligonucleotide probes to granular gland cell and spirocysts (Fig. 3D).

Figure 3 The application of common approaches designed to ameliorate non-specific fluorescence in situ hybridization (FISH) probe binding failed to inhibit hybridization to granular gland cells (Gc) and spirocysts (Sp).

Cy3 (A), Cy5-labelled NonEUB338 (B) and Atto 647-labelled Vib-GV (C) FISH probes all hybridized to spirocysts. Blocking solution also failed to prevent non-specific binding of oligonucleotide probes to granular gland cells (D). Scale bars represent 100 µm in (A)–(C) and 50 µm in (D). Abbreviations: Gc, granular gland cell; Sp, spirocysts; and Symb, Symbiodinium.

Difficulties discriminating true detection of bacterial cells from non-specific binding to non-target cellular structures is demonstrated in tissue sections of Acropora hyacinthus affected by a WS disease (Fig. 2H). Coral-associated microbial aggregates (CAMAs) can be observed within the calicoblastic layer of healthy tissues sampled from WS diseased colonies (n = 3) and probed with the EUB338 probe (Fig. 2H). CAMAs have been identified in healthy tissues of many coral species (Work & Aeby, 2014), although their exact functional role is unknown and their influence on a coral’s position along a healthy-diseased continuum needs further exploration. Non-specific binding to granular gland cells was also observed within gastrodermal cells that were in the vicinity of a bacterial aggregate, and could be mistaken for a true positive without detailed understanding of coral cellular structures (Fig. 2H). True positive detection of bacterial assemblages and individual bacterial cells were observed in all regions (n = 6) of sections from actively progression WS lesions (Fig. 2I).

Recommendations to avoid common pitfalls

The inability to prevent non-specific binding of oligonucleotide probes to cellular structures within coral tissues represents a major challenge for accurate identification of specific bacterial targets. However, by taking into account the issues highlighted above and the recommendations laid out below, these challenges can be overcome.

• Combining appropriate microscope hardware with advanced image acquisition software significantly improves detection of bacterial cells within coral tissues. Laser confocal scanning microscopes (LSCM) provide the ideal platform for optimized image resolution, fluorescence signal acquisition, and removal of non-specific background auto-fluorescence. While standard fluorescence microscopes can be used to visualize coral samples for some applications (e.g., coral mucus-associated microorganisms), high sample auto-fluorescence often confounds coral FISH studies. Unfortunately, LSCMs are relatively expensive, which limits access for some researchers, but many of the suggestions outlined below should prove helpful, regardless of the microscope/software available.

• Selecting an appropriate fluorochrome with an emission spectrum distinct from that of coral tissues (e.g., background green fluorescent proteins) and Symbiodinium cells (i.e., background fluorescence in red and far red due to the presence of chlorophyll) is an important consideration (Ainsworth et al., 2006a). Certain fluorochromes (e.g., FITC) are not ideal for visualizing targets within coral tissues, while others (i.e., Cy3, Cy5 and Atto647) are commonly used for coral samples. However, it is important to note that few, if any, fluorochromes provide emission spectra completely distinct from that of coral tissue and Symbiodinium auto-fluorescence.

• When using microscopes that allow spectral profiling at the pixel scale, comparison of background emission spectra and intensity with that of potential positive signals (i.e., probe binding) can greatly improve differentiation between background fluorescence and true probe binding.

• Alternative approaches to FISH, such as catalyzed reporter deposition FISH (CARD-FISH) that which increase probe emission signal intensity, can aid in the differentiation of target versus non-target fluorescence. Such approaches have been widely used in other environmental samples (Pernthaler, Pernthaler & Amann, 2002) and are now being successfully used for corals (see examples in Ainsworth et al., 2006a; Bayer et al., 2013; Neave et al., 2016).

• Serial sections should be visualized using both FISH and traditional staining (e.g., H&E stain or Trichrome stains) to identify and localize granular gland cells within the gastrodermal cell layer, particularly within the glandular band of mesenterial filaments, and spirocysts within epidermal cell layers, particularly in coenosarc and stomodeum regions of the polyp.

• A nonsense probe representing a true negative control must be applied to serial tissue sections (or on the same section) to help distinguish specific probe binding to target bacterial populations from non-specific probe binding to coral cellular structures (Wallner, Amann & Beisker, 1993).

• Familiarity with coral tissue structures is imperative to distinguish morphological characteristics, such as size and shape that differentiate bacteria from coral tissue structures, including spirocysts and granular gland cells that are prone to non-specific probe binding (Posch et al., 1997; Posch et al., 2009). Undertaking practical courses that focus on histological analysis of cellular structures, particularly focused on coral tissues, can aid in the familiarization process, along with close collaborative networks with practitioners already trained in this area.

Non-specific probe binding constitutes a significant impediment to FISH-based detection of coral-associated bacterial communities. However, the inclusion of appropriate control measures combined with observational rigor can greatly increase the ability of practitioners to accurately distinguish non-specific probe binding and host auto-fluorescence from true positive bacterial signals. Detailed approaches for the application of FISH to both healthy and diseased corals are summarized in Fig. S1 and further details of pitfalls and possible solutions are summarized in Table S1. Following these recommendations will allow practitioners to avoid confusion associated with non-specific binding and allow FISH techniques to be employed to their full potential in studies of coral tissues.

Supplemental Information

Figure S1 Schematic flow chart detailing the protocol to process coral samples for both histological and FISH based visualization of coral tissue structure and identification of bacterial communities associated with coral tissues

Click here for additional data file.

Table S1 Pitfalls and solutions to overcome FISH challenges

Identified pitfalls and potential solutions to overcome challenges of applying FISH approaches to coral tissues.

Click here for additional data file.

Data S1 Raw data associated with FISH investigations of coral tissues from the study

Click here for additional data file.

The authors would like to thank Dr. Jean-Baptiste Raina and Dr. Yui Sato for their logistical help during the project.

Additional Information and Declarations

Competing Interests

Author Contributions

Field Study Permissions

Data Availability

The authors declare there are no competing interests.

Naohisa Wada and Frederic J. Pollock conceived and designed the experiments, performed the experiments, analyzed the data, wrote the paper, prepared figures and/or tables, reviewed drafts of the paper.

Bette L. Willis conceived and designed the experiments, contributed reagents/materials/analysis tools, wrote the paper, reviewed drafts of the paper.

Tracy Ainsworth and Nobuhiro Mano contributed reagents/materials/analysis tools, wrote the paper, reviewed drafts of the paper.

David G. Bourne conceived and designed the experiments, analyzed the data, contributed reagents/materials/analysis tools, wrote the paper, prepared figures and/or tables, reviewed drafts of the paper.

The following information was supplied relating to field study approvals (i.e., approving body and any reference numbers):

Sampling was undertaken on Permit G11/34003.1 issued by the Great Barrier Marine Park Authority.

The following information was supplied regarding data availability:

The raw data is included in the manuscript.

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
