# Peer review of "In situ visualization of bacterial populations in coral tissues: pitfalls and solutions"

_PeerJ, doi:10.7717/peerj.2424_

## Round 0.1 · original submission · Minor Revisions

· Academic Editor

Minor Revisions

Dear Dr. David Bourne
There are some minor revisions required.
In addition to the two referees remarks, please inform the number of coral specimens they sampled and used in their FISH assays, and an explanation for the choice of a probe hybridization temperature of 46oC and any optimization they might have made, since this is critical for probe specificity. I will look forward to listen from you. Fabiano

Reviewer 1 ·

Basic reporting

Wada et al. provide valuable information regarding coral tissue microbial community/assembly visualization in the manuscript “In situ visualization of bacterial populations in coral tissues: pitfalls and solutions”. The manuscript is relevant and well written and should be published after minor revisions. Please see bellow my comments, I hope authors find it useful to improve their manuscript.

Experimental design

Despite authors have multiple samples from different coral colonies is not clear enough how they treated them. Considering this, there are some statistical analysis that could support authors ideas/results?

I suggest including a workflow flowchart illustrating all the steps to achieve the results (manuscript figures).

If authors and Editor find appropriate, I also suggest authors include a step-by-step protocol as Supplementary Information. In this material solution receipts and preparation recommendations, for instance, would benefit readers to follow authors’ methods much better.

Validity of the findings

Authors collected many samples from diseased and healthy corals but are not clear both in Material and Methods and Results and discussion sections how they analyzed it quantitatively and how robust their approach is. Authors need to make this information clear in manuscript.

Additional comments

Abstract:

I suggest including more information regarding the results obtained in the present study as well describe better the conclusive remarks made in the main text body.

Line 28-33: This sentence is too long. Please revise to improve readability.

Introduction

Line 50: Authors should carefully check the references before resubmitting. Moter & Gobel, 2000 do not have reference to J Microbiol Methods, for instance.

Line 58: What would be these interactions?

Line 63: Endozoicomonas?

Line 70: Authors would explain briefly what is coral auto-fluorescence in introduction. Maybe just one sentence would be enough.

Material and Methods

Line 80: Would authors include a picture of both healthy and WS-infected corals? I understand that “WS” is White Syndrome, right? Please do not abbreviate here. This is the first time to mention WS.

Line 84: How many samples in total? How many colonies? Please make sure to describe this information clearly here.

Line 94: Please write the full term of “PBS” before abbreviating it. For how many time and in what temperature did authors embedded samples in 1.5% agarose?

Line 98: How many days in total?

Line 107: Did authors develop all these methods? If so, please make sure to make it clear here.

Line 110: Manufacture for Mayer’s Heamalum? Please add information regarding what structures/cells this reagent attaches to.

Line 118: Did authors develop all these methods? If so, please make sure to make it clear here.

Line 150: Please provide better details about what were the problems to be overcome in this subsection.

Line 151: this sentence is not clear. Please revise it.

Results and Discussion

Lines 166-168: I think this information would be moved to Methods section.

Lines 193-197: I understand that analyzing all those microscopies is a very laboring task and ones need to be well trained but how? Maybe authors would provide readers valuable information regarding how someone can be prepared to be familiar with coral cellular structure and differentiate it. How to avoid the intrinsic subjective interpretation?

Lines 215-216: What would be “detailed understanding” objectively? Please the see comment above.


Figures

Figures quality is very good. I suggest moving figure S1 to the main text.

If authors and editor find appropriate, I would suggest including a flowchart representing all the workflow.

Reviewer 2 ·

Basic reporting

The ms describes a series of improvements in FISH techniques to allow unambiguous detection of bacteria in coral tissues, mainly to overcome the problem of non-specific binding of targeted rRNA probes to cellular structures within the coral tissues. The work done is scientifically sound and the text is very well written. Methods, to the best of my knowledge, are appropriate. Results are clearly presented with high quality micrographs. I have only some minor comments that I list below.

line 80: what is WS-infected? White Syndrome? Please explain.

line 84: how many specimens were sampled?

line 123: did you mean proteinase K?

lines 131: how was the probe temperature of 46C chosen? Did you do any optimization? If so please explain.

line 205. Why non-fluorescent? Shouldn't it be "fluorescent" or "non-specific signal" or "auto-fluorescence"?

line 255: versus

Experimental design

no comments

Validity of the findings

no comments

Additional comments

no comments

---

## Round 0.2 · accepted · Accept

· Academic Editor

Accept

Dear Dr. Bourne. Congratulations, your article is Accepted.

Sincerely. Fabiano Thompson.